# Treating Anti-Vax Patients, a New Occupational Stressor—Data from the 4th Wave of the Prospective Study of Intensivists and COVID-19 (PSIC)

**DOI:** 10.3390/ijerph19105889

**Published:** 2022-05-12

**Authors:** Nicola Magnavita, Paolo Maurizio Soave, Massimo Antonelli

**Affiliations:** 1Postgraduate School of Occupational Medicine, Università Cattolica del Sacro Cuore, 00168 Rome, Italy; paolomaurizio.soave@policlinicogemelli.it; 2Department of Woman, Child and Public Health Sciences, Fondazione Policlinico Universitario Agostino Gemelli IRCCS, 00168 Rome, Italy; 3Department of Emergency, Anesthesiology and Resuscitation Sciences, Fondazione Policlinico Universitario Agostino Gemelli IRCCS, 00168 Rome, Italy; massimo.antonelli@unicatt.it

**Keywords:** longitudinal study, emergency, infectious disease, organizational justice, stress, loneliness, compassion fatigue, meditation, prayer, insomnia, mental health, anesthetists, occupational health

## Abstract

The Prospective Study of Intensivists and COVID-19 (PSIC) is a longitudinal study that besides investigating a cohort of intensivists from one of the two COVID-19 hub hospitals in Central Italy since the beginning of the pandemic (first wave, April 2020), has conducted a new survey at each successive wave. In addition to the variables investigated in previous surveys (job changes due to the pandemic, justice of safety procedures, job stress, sleep quality, satisfaction, happiness, anxiety, depression, burnout, and intention to quit), the latest fourth wave (December 2021) study has evaluated discomfort in caring for anti-vax patients. A multivariate logistic regression model confirmed that high levels of occupational stress (distressed 75.8%) were associated with isolation, monotony, lack of time for meditation, and poor relationships with anti-vaccination patients. Compared to the first phase, there was a reduction in levels of insomnia and anxiety, but the percentage of intensivists manifesting symptoms of depression remained high (58.9%). The study underlined the efficacy of organizational interventions and psychological support.

## 1. Introduction

Since the beginning of the pandemic, intensive care physicians (ICPs) providing care for patients with coronavirus-19 (COVID-19), caused by the severe acute respiratory syndrome coronavirus 2 (SARS-CoV-2), have been exposed to a number of occupational stressors. In the first phase of the pandemic, lack of knowledge about the disease, poor environmental prevention measures and inadequate personal protective equipment were the main stressors for healthcare workers [1]. Occupational stress was associated with lack of information on safety [2], fear of the onset of symptoms, or positivity following RT/PCR tests [3]. In consideration of this situation, in April 2020, we launched a prospective study of the intensivists involved in the treatment of COVID-19 patients in one of the two hub centers in Central Italy, with the intention of assessing their mental health in relation to the different phases of the pandemic. The Prospective Study of Intensivists and COVID-19 (PSIC) has continued so far, with a new survey corresponding to each pandemic wave.

Gradually, a better understanding of the new disease and improved safety procedures overcame fear of the unknown, but ICPs in COVID-19 centers continued to have a high workload that prevented them from finding relief from stress through meditation or physical activity. The need to inform relatives of the adverse outcome of therapy increased compassion fatigue. Furthermore, work in COVID-19 patient units became increasingly monotonous, and isolation not only in the workplace but also in daily life became increasingly burdensome. After the first months of the pandemic, the highly favorable attitude of the public towards ICPs changed, and there was a resumption of malpractice complaints against doctors [4]. At the end of 2020, after the second epidemic wave [5], disorders such as depression and burnout occurred among the ICPs [6].

Although intensivists benefited from an easing of isolation restrictions and a partial resumption of social activities following the introduction and spread of vaccines, COVID-19 units went on being inundated with patients during the third and fourth waves, and the work of the intensivists continued with the same intensity. There was, however, an important innovation: most of the hospitalized patients had refused vaccination. Many of them were extremely hostile towards the medical profession and, at times, determined to refuse treatment.

In this fourth survey of the PSIC we continued the analysis of the changes in the mental health of doctors who had been treating COVID-19 patients since the beginning of the pandemic, in relation to changing environmental and working conditions. Moreover, we wanted to evaluate the extent to which the relationship with anti-vax patients generated occupational stress in intensivists.

## 2. Materials and Methods

### 2.1. Participants

This prospective study, which was designed to investigate ICPs during the successive pandemic waves, began in April 2020, and involved anesthetists working in one of the two COVID-19 hub hospitals in Central Italy. Repeated baseline surveys were carried out to correspond with the first, second (December 2020), third (April 2021), and recent fourth wave of the pandemic (December 2021). On this occasion, as in the previous ones, the ICPs were confidentially contacted by email and asked to participate through the SurveyMonkey© online platform. The answers were collected anonymously on a specific file without any individual reference. Participation was completely voluntary, and no economic incentive was provided for response. Two weeks after commencing the investigation, a reminder email was sent with the preliminary results of the study. The ICPs were informed both on the results collected in the various surveys and the findings published in scientific journals.

Since the cohort was mobile and the survey was anonymous, the prospective study had a repeated cross-sectional character, which allows a comparison of the point prevalence measured in the various surveys but does not allow for the calculation of the incidence.

Over the course of the study, the total number of ICPs on duty in the COVID-19 hospital ranged from 182 (1st wave) to 205 (2nd wave) and 198 (3rd and 4th waves). In all surveys, the participants constituted a significant share of those eligible.

The survey was conducted in accordance with the Helsinki Declaration. The Catholic University Ethics Committee approved the study (ID 3292; 30 September 2020).

### 2.2. Questionnaire

The questionnaire included a series of ad hoc questions on occupational variations in relation to the pandemic, and a panel of standardized questionnaires measuring occupational and emotional outcomes.

The questions in the ad hoc part of the questionnaire were obtained prior to the start of the survey through structured interviews in a focus group conducted on a small group of anesthetists. Before each new data collection, the authors discussed to decide if there was the possibility of reducing the number of questions, to favor the answer and reduce the loss of observations which is a frequent problem in longitudinal studies.

The ICPs were asked to indicate the extent of their workload in the current situation compared to the past, by choosing one of five responses ranging from “much less than usual” to “much greater than usual”. Similarly, they were required to indicate (with the same 5-point Likert-type scale) how much time they were spending on physical activity and, respectively, meditation, prayer, or spiritual/mental activities, compared to what they did before the epidemic.

Furthermore, they were asked to indicate whether they agreed that during the epidemic their work had become more monotonous and repetitive; that the task of informing relatives of the death of a patient had been more frequent; that at work they were increasingly isolated; and that in social life they were increasingly isolated. In each of these questions, they could choose the answer from a 5-point Likert-type scale ranging from “I strongly disagree” to “I absolutely agree”.

Procedural justice perceived in safety measures was measured with the Italian version [7] of the Colquitt questionnaire [8,9,10] which is composed of 3 items (e.g., “Are these procedures error-free?”). Each question was answered according to a 5-point Likert scale, from 1 = “I totally disagree” to 5 = “I strongly agree”; thus producing a scale ranging from 3 to 15. In this 4th survey, the reliability of the questionnaire, measured by Cronbach’s alpha, was 0.660 (acceptable).

Work stress was assessed with the Italian version [11,12] of the Siegrist effort/reward imbalance model [13,14]. The questionnaire consisted of 10 items with responses ranging on a 4-point Likert scale from “1 = strongly disagree” to “4 = strongly agree”. The effort subscale contained three questions (e.g., “My job has become more and more demanding”); the total score ranged from 3 to 12. The reward sub-scale was based on seven questions (e.g., “I receive the respect I deserve from my superior or an equivalently qualified person”); consequently, this score ranged from 7 to 28. Internal consistency reliability in this survey was 0.798 for effort (very good) and 0.823 for reward (very good).

Sleep quality was determined using the 2-item version of the “Sleep Condition Indicator” (SCI-02) [15,16], a brief scale that evaluates insomnia disorder in everyday clinical practice, according to the Diagnostic Statistic Manual 5 (DSM5). Each question (e.g., “How many nights a week have you had a problem with your sleep during the past month?”) was graded on a 5-point Likert scale, ranging from 4 to 0. The final score ranged between 0 and 8, with higher values indicating better sleep quality. Cronbach’s alpha was 0.759 (good). A score of ≤4 revealed possible insomnia disorder.

Mental health was measured on Goldberg’s anxiety and depression scales (GADS) [17,18], composed of 18 binary items on anxiety (9 items) and depression (9 items). Typical questions were: “Have you had difficulty relaxing?” for anxiety, and “Have you lost confidence in yourself?” for depression.

Other questionnaires were: job satisfaction, expressed according to Warr et al. [19,20] by a single question on a 7-point Likert scale ranging from extremely dissatisfied to extremely satisfied; happiness, measured by the 10-point Ab-del-Khalek’s single item scale [21]; burnout feelings, evaluated according to West et al. [22] on a 6-point scale; and the intention to quit the hospital, measured with a single item (yes/no). A detailed description of the questionnaires and their psychometric properties has been published in previous reports [6,23,24]. In this fourth survey, the relationship with patients belonging to the anti-vaccination movement was investigated using the Italian version [25] of the “contact with patients and their family” sub-scale of the Nurses Work Functioning Questionnaire NWFQ [26]. This scale is made up of 8 questions, with answers graded from 0 to 6; the score therefore varies between 0 and 48. An example of a question is: “In the last 4 weeks, how much difficulty have you had in interacting either with patients who refuse the anti-COVID-19 vaccination or with their relatives/careers?”. The reliability of the questionnaire (Cronbach’s alpha) in this study was 0.875.

### 2.3. Statistics

The distribution of the variables was analyzed by measuring central tendency (mean, median, mode) and dispersion (standard deviation). According to the repeated cross-sectional design, we calculated the punctual frequency of the variables of interest and compared this prevalence of the 4th survey with those collected in the previous pandemic phases by the chi-square test for categorical data or by ANOVA and post-hoc comparison according to Bonferroni for continuous variables.

We therefore wanted to understand which were the most important occupational stressors in the 4th phase. The effect of changes in work patterns on occupational stress was studied using multiple logistic regression, in which age, gender, and all changes were included as predictors, and ERI, dichotomized using 1 as the cut-off, was entered as the dependent variable. In this way, it was possible to calculate the adjusted odds ratio and the 95% confidence intervals for each of the pandemic changes.

Similarly, the effect on health outcomes due to perception of occupational justice, effort, and reward was studied using multiple logistic regression in which age and gender were postulated as confounders and the three work-related variables as predictors.

Analyses were performed using IBM/SPSS 26.0 (IBM Corporation, Armonk, NY, USA).

## 3. Results

### 3.1. Characteristics of the Participants

Of the 198 eligible workers who were in service on 1 December 2021, 110 participated in the survey (participation rate = 55.6%). Fifteen gave incomplete answers and were therefore excluded from subsequent analyses. This high number of interruptions before the conclusion of the questionnaire was a new phenomenon; in previous surveys there had been only a few sporadic cases. In intensivists, the interruption of a questionnaire may depend on the need to respond to a medical emergency. In this case, we fear that one reason might be the boredom of answering the same questions for the fourth time. The participants were mainly young (61.1% under 35 years of age), female (55.5%), and had been employed in the hospital for more than three years (67.4%). Most of them were specialists and had a permanent work contract (55.8%), while the others were residents who had been hired with temporary contracts during the COVID-19 emergency.

### 3.2. Features of the 4th Phase and Comparison with the Previous Surveys

In Table 1, the characteristics of the sample in this fourth survey are compared with those of the previous phases of the prospective study. The proportion of workers who reported unprotected exposure to COVID-19 patients (44.2%) was significantly higher in this fourth survey than at baseline (24.7%) but was significantly lower than in the two previous surveys (Table 1). Surprisingly, compared to previous surveys, many unprotected contacts had occurred exclusively at home (32.6%), or both at home and at work (48.4%). Only 26.3% of ICPs reporting an unprotected contact attributed it to a hospitalized patient, while in the previous survey this percentage was 78.9%. Nineteen ICPs (20.0%) reported having contracted COVID-19; 14 of them (73.7%) were infected during the 1st and 2nd waves, before being vaccinated; the remainder contracted COVID-19 after a complete course of vaccinations. In most cases (94.7%), infection was completely asymptomatic (15.8%) or resulted in mild symptoms that did not require treatment (78.9%). However, a significant proportion of the subjects who had contracted the disease reported protracted symptoms (long-COVID-19, 47.4%) or permanent outcomes (post-COVID-19, 5.3%) after the infectious phase (Table 1).

### 3.3. Causes of Occupational Stress during the 4th Phase

The impact of changes in work patterns on occupational stress perceived during the 4th survey was investigated by multivariate logistic regression (Table 2). In addition to social isolation, monotony resulting from having to treat only one type of patient with rigidly standardized protocols and lack of time for meditation, prayer, and relaxation, a further factor arising from difficult relationships with patients belonging to the anti-vaccination movement (and with their families) was significantly associated with occupational stress (Table 2). Multivariate logistic regression expressed a consistent percentage of the variance in perceived stress, thus indicating the efficacy of the model. In the various groups of participants, the Mann–Whitney U test found no difference in the degree of discomfort experienced in interaction with anti-vax patients in relation to gender, age group, or occupational status.

### 3.4. Changes in Occupational Stressors and Mental Health Effects during the Pandemic

A comparison of the responses received during the four waves (Table 3) showed that workload, isolation, and compassion fatigue resulting from the need to inform relatives of an unfavorable outcome of treatment, increased significantly during the two years of the pandemic. The percentage of distressed ICPs did not change significantly during the observation period, while findings indicated a significant reduction in the number of participants with sleep problems, a non-significant reduction in ICPs reporting anxiety, and an increase in subjects affected by symptoms of depression (Table 3).

Confidence in safety measures increased significantly over the period with a parallel rise in the quality of sleep, as demonstrated by the Bonferroni test which was used to compare the fourth and first waves. Levels of occupational effort and depression increased significantly over the period, with no significant changes between the current survey and the previous one (Table 4).

### 3.5. Association between Occupational Stress and Mental Health during the 4th Phase

The associations between the variables expressing work discomfort (procedural justice, effort, reward) and health outcomes were much weaker than those observed in the previous surveys. Occupational rewards protected against symptoms of depression and increased job satisfaction associated with the perception of fairness in safety procedures. The risk of burnout was associated with work effort. Anxiety and happiness scores and intent to quit were not determined by occupational stressors at this stage of the pandemic (Table 5).

## 4. Discussion

In the two years since the pandemic began, frontline healthcare workers have had to face an unprecedented series of stressors. This study, which, to the best of our knowledge, is the only longitudinal study of intensive care doctors dealing with COVID-19 patients, has enabled us to observe the evolution of workers’ mental health in the different phases of the pandemic.

The mental health of healthcare workers during the pandemic is a topic that has given rise to numerous publications. On 1 March 2022, the search phrase “COVID-19 and (health care workers) and (mental health or anxiety or depression or burnout or stress or sleep problems or insomnia)” produced 2807 articles on PubMed, 88 of which were systematic reviews or meta-analyses. Unfortunately, most of these studies considered the pandemic to be a precise phenomenon, while it is clear (as our study confirms) that the epidemic has modified occupational stress factors continuously and consequently also their effect on workers’ health. Review studies on frontline healthcare workers have reported high rates of post-traumatic stress disorder [27,28,29,30], anxiety [31,32,33,34], sleep problems [35,36,37], depression [38,39,40,41], and burnout [42,43,44]. However, these studies could be biased since they do not take into account the fact that the original observations were made at different times. Furthermore, mere observation of a mental health problem during an epidemic does not mean its cause can be attributed to the pandemic. In the case of COVID-19, there are no pre–post studies that have compared the mental health level of workers during the pandemic with previous measurements, nor are there currently any longitudinal studies that assess the evolution of workers’ mental health during the pandemic. Ours is the first prospective study that has followed a group of ICPs constantly engaged in the treatment of COVID-19 patients in one of the two COVID-19 hub hospitals in Central Italy. It has enabled us to observe how mental health has changed in relation to the different phases of the pandemic.

The first phase, characterized by fear of an unknown epidemic and uncertainty about safety measures, witnessed a high level of sleep problems and anxiety in ICPs [23]. Sleep problems have gradually decreased, and two years after the outbreak of the pandemic, anxiety symptoms are less frequent than previously observed. Current levels of sleep quality may no longer express the fear that legitimately manifested itself during the first phase of the pandemic but may be linked to night work and the tension that occupational tasks induce in anesthetists. The prolonged pandemic and constant work pressure, together with high workload, isolation, compassion fatigue, and less time for meditation, have resulted in an increase in depression, burnout, and intention to quit. These factors were already apparent at the beginning of the pandemic in ICPs [6,24], and still persist at the current time.

Recently, a further problem has been added to this complex situation. Over time, the public attitude towards doctors has changed dramatically: for some, instead of being heroes, doctors have become guilty parties. This has caused an increase in litigation against doctors [44] and has also led to unscientific theories being pitted against medical ones. Intensive care units are currently occupied mainly by patients who refuse vaccinations, fear being secretly vaccinated against their will, or in some cases, vehemently reject all medical practices. Under these conditions it is very difficult to establish the climate of empathy that is so important for patients undergoing treatment, and this gives rise to much of the occupational stress perceived by intensivists dealing with the fourth phase. They have learned new therapeutic techniques and safety measures that allow them to face the disease with determination rather than fear, even though a fifth of them reported contracting an infection which, in 50% of cases, has led to prolonged or persistent symptoms. Caring for anti-vaccination patients has added a new occupational stressor to a context already fraught with problems.

Diffidence towards vaccines is not a new phenomenon. In some cases, medical mistrust reflects very real historical and on-going injustices experienced by socially and economically marginalized groups [45]. However, it often arises from the so-called Dunning–Kruger effect in subjects whose lack of knowledge and expertise leads them to have an overrated perception of their own competence. The consequent misperception can then lead to a paranoid state in which vaccines are seen as a toxic bio-weapon inflicted by ruthless authoritarians with persecutory intent [46]. If the perception of a threat intensifies, paranoia can easily induce aggression [47]. For this reason, some anti-vax patients react in a threatening and aggressive way towards the medical staff treating them. In recent months, the Italian national press has reported an increase in the number of people who reject anti-COVID-19 treatment [48,49,50,51,52,53,54] and react aggressively towards healthcare workers [55,56,57,58,59,60]. Political factors have often favored the radicalization of positions [61], as has the circulation of fake news on social media. The strong suspicion that the anti-vax phenomenon is supported by political organizations or foreign countries has led some to request coordinated intervention by cyber security and anti-terrorism services [62]. Others suggest adopting a more positive attitude towards the anti-vaccination movement in order to establish a constructive dialogue [63], or even view it as a different kind of religion [64]. Clearly, occupational and compassionate skills alone are insufficient to deal with such a complex problem. Since hospitals cannot run the risk of endangering their staff, health companies should issue a clear policy outlining behavioral guidelines for staff and indicating ways of informing patients that their autonomy will not be undermined by the administration of drugs or vaccines without their consent, and that all treatments offered are strictly necessary for their health and survival.

The stress model we investigated referred exclusively to work factors that have changed during the pandemic. We did not consider the intrinsic stressors in the work of anesthesiologist and more generally those of which no variation has been reported to us. Taking this into consideration, the model shown in Table 2 appears to effectively explain the observed increase in distress cases.

In correspondence with the fourth pandemic wave, the increase in cases of distress was mainly due to monotony, isolation, difficulty in relating with anti-vax patients, and lack of time for meditation. In previous investigations, others were the dominant stressors. At baseline, at the first stage, stress was mainly driven by unprotected exposure to COVID-19 positive patients and lack of confidence in safety measures; compassion fatigue was nearly significant [23]. These findings perfectly expressed the situation of the first pandemic phase, in which the ICPs had to deal with an unknown disease, against which there was a lack of safety measures and therapies. The situation has gradually changed. After the second wave, the most relevant stressors, besides the distrust in security measures and excessive workload, were the lack of time for meditation and isolation [6]. Working alone and with little confidence in safety measures were the most significant predictors of distress in the third wave [24]. Confidence in safety measures is not a major stressor today, probably because ICPs have assimilated the safety procedures imposed by the pandemic and observed that COVID-19 infection often originates from non-occupational exposure.

The risk factors that varied most over the observation period were workload, compassion fatigue, and isolation. Factors significantly associated with occupational stress were monotony, isolation, lack of time for meditation, and the relationship with anti-vax patients. These results were consistent with expectations and previous observations in the literature. A number of authors [65,66,67] have reported the association between excessive workload and stress in medical staff during the COVID-19 pandemic. Isolation has often been associated with anxiety and depression symptoms [68,69], and compassion fatigue has been associated with secondary traumatic stress [70]. Meditation has proven to be a stress-reducing factor in occupational cohorts [71,72], while monotony has so far been studied mainly in patients [73]. The issue of the relationship with patients who reject vaccines has not yet been addressed in other studies.

In this fourth wave, occupational stress has a modest association with depression, burnout, job satisfaction, and intention to stay, and no relation with anxiety and happiness. A few months ago, during the third phase, occupational effort and lack of rewards were very significantly associated with anxiety, depression, burnout, job dissatisfaction, and intention to quit [24]. This attenuation of the link between occupational stress caused by the pandemic and negative emotional outcomes could indicate that ICPs are moving from emergency to the routine management of an endemic clinical problem.

Acute distress is turning into chronic distress. The prolonged state of suffering of medical staff dealing with COVID-19 is alarming and requires intervention to alleviate some of its causes. Additional ICPs could be taken on to lighten the workload, but this measure is hampered by a shortage of qualified staff. Lack of time for intellectual activities, meditation or prayer also contributes to the excessive workload. To prevent monotony anesthesiologists could be assigned to tasks that do not always involve caring for COVID-19 patients. Improvement in therapeutic procedures could indirectly alleviate compassion fatigue but, in the meantime, doctors undoubtedly need psychological support.

A strong point of this study is the prospective method used to observe the evolution of the pandemic and workers’ responses since it enabled us to understand which mental health problems were the result of the pandemic and which were an intrinsic part of the intensivists’ work. Moreover, by investigating the relationship with patients adhering to the anti-vaccination movement, it is the first study to consider this new and important occupational risk factor for physicians. However, this survey also has some limitations: Since answers were anonymous, we were able to calculate only the current prevalence, but not the incidence of the phenomena in relation to the various measurements. Moreover, by selecting a single hub hospital we were unable to extend our findings to other experiences, although there is no reason to conclude that the perceptions of other doctors engaged in the pandemic differ from those described above.

## 5. Conclusions

Excessively high levels of workload together with isolation, monotony, compassion fatigue, lack of time for meditation, and the recent additional need to interact with anti-vax patients have resulted in two years of chronic stress for doctors caring for COVID-19 patients. Although sleep problems and anxiety seem to have improved, levels of perceived stress and depression are still extremely high. This situation should prompt healthcare managers to focus their attention on the crucial role performed by these workers in providing quality healthcare in our society [74]. Structural and administrative measures that incorporate rational planning of staff and timetables should be adopted in order to prevent workers from incurring mental health risks. It is also important to create an effective safety system (preferably of a participatory nature) that focuses particularly on specific occupational training. Shared policies are needed in order to effectively manage patients belonging to the anti-vax movement. Individual psychological support and the provision of relaxation techniques can help physicians overcome stress associated with compassion fatigue, patient relationships, and other occupational stressors.

## Figures and Tables

**Table 1 ijerph-19-05889-t001:** Characteristics of the population in the different phases of the prospective study.

Variables	Baseline		2nd Survey		3rd Survey		4th Survey		X^2^
	*N*	%	*N*	%	*N*	%	*N*	%	*p*
Eligible	182		205		198		198		
Participant (% eligible)	154	84.6	105	51.2	120	60.6	95	48.0	-
Resident (% participant)	58	37.7	55	52.4	68	56.7	42	44.2	0.010
Gender, male (% participant)	75	48.7	51	48.6	58	48.3	47	49.5	0.999
Age, <35 years (% participant)	94	61.0	76	72.4	84	70.0	58	61.1	0.141
Reporting unprotected exposure to COVID-19 patients	38	24.7	59	56.2	71	59.2	42	44.2	0.000
Reporting COVID-19 disease (% participant)	-	-	16	15.2	26	21.7	19	20.0	0.637
Asymptomatic COVID-19 case (% cases)	-	-	6	37.5	10	38.5	3	15.8	0.488
Mild COVID-19 case (% cases)	-	-	9	56.3	14	53.8	15	78.9	
Moderate COVID-19 case (% cases)	-	-	1	6.3	2	7.7	1	5.3	
Reporting long-COVID-19 (% cases)	-	-	-	-	10	38.5	9	47.4	-
Reporting post-COVID-19 (% cases)	-	-	-	-	1	3.8	1	5.3	-

**Table 2 ijerph-19-05889-t002:** Impact of changes in work patterns on occupational stress (ERI > 1) during the 4th pandemic phase. Odds ratios (aOR) and Confidence Intervals at 95% (CI 95%) adjusted for age and sex: 95 cases.

Variable (Mean ± sd)	OR	CI 95%	p
Workload (4.27 ± 0.63)	1.343	0.248; 7.287	0.733
Physical activity (1.98 ± 0.97)	0.876	0.415; 1.847	0.727
Meditation (2.17 ± 0.99)	0.335	0.129; 0.869	0.025
Monotony (3.01 ± 1.28)	2.703	1.065; 6.855	0.036
Compassion fatigue (3.73 ± 1.05)	1.221	0.514; 2.903	0.651
Working alone (2.74 ± 1.18)	0.522	0.183; 1.486	0.223
Isolation (3.19 ± 1.25)	3.730	1.381; 10.070	0.009
Procedural justice (8.72 ± 2.15)	1.156	0.778; 1.719	0.473
Relationship with anti-vax (20.83 ± 11.66)	1.197	1.061; 1.350	0.004
Determination coefficient (Nagelkerke R^2^)	0.679		

**Table 3 ijerph-19-05889-t003:** Changes reported during the COVID-19 outbreak and prevalence of high stress, insomnia, anxiety, and depression during the four pandemic waves.

Reported Effect	Baseline		2nd Survey		3rd Survey		4th Survey		X^2^
	*N*	%	*N*	%	*N*	%	*N*	%	*p*
Increased/greatly increased workload	77	50.0	83	83.0	98	84.5	86	81.5	0.000
The work became more repetitive and monotonous	51	33.1	36	36.0	53	45.7	40	42.1	0.352
More frequent need to inform of the death of a relative	61	39.6	65	65.0	81	69.8	67	70.5	0.000
Isolation at work			42	42.0	47	40.5	30	31,6	0.686
Isolation in daily life			81	81.0	78	67.2	47	49.5	0.001
Time for physical exercise was shorter/much shorter	117	76.0	80	80.0	92	79.3	71	74.7	0.486
Time for meditation was shorter/much shorter	72	46.8	65	65.0	74	63.8	55	57.9	0.100
Distressed (effort/reward weighted ratio > 1)	117	76.0	80	80.0	83	72.8	72	75.8	0.678
Insomniac (SCI08 score ≤ 16; SCI02 score ≤ 4)	63	40.9	33	33.0	32	28.1	23	24.2	0.030
Anxious (GADS anxiety score ≥ 5)	40	26.0	31	31.0	29	25.4	18	18.9	0.289
Depressed (GADS depression score ≥ 2)	75	48.7	63	63.0	73	64.0	56	58.9	0.042

SCI08 = Sleep Condition Indicator, used in the baseline survey; SCI02 = Sleep Condition Indicator, short form, two items; used in the 2nd, 3rd, and 4th survey; GADS = Goldberg anxiety and depression scale.

**Table 4 ijerph-19-05889-t004:** Mental health indicators (perceived justice, occupational stress, sleep quality, anxiety, depression) in anesthesiologists during the COVID-19 pandemic waves.

Variable	1st Wave	2nd Wave	3rd Wave	4th Wave	ANOVA	Bonferroni		
	Mean ± sd	Mean ± sd	Mean ± sd	Mean ± sd	*p*	*p*		
						4 vs. 1	4 vs.2	4 vs. 3
Procedural justice	49.91 ± 13.64	53.60 ± 15.80	53.33 ± 15.67	58.10 ± 14.35	0.001	0.001	0.203	0.123
Effort	8.56 ± 1.95	9.35 ± 1.68	9.28 ± 1.74	9.09 ± 2.00	0.002	0.164	1.000	1.000
Reward	16.49 ± 3.68	16.62 ± 3.91	17.19 ± 3.91	16.44 ± 4.15	0.427			
Job stress	1.30 ± 0.51	1.42 ± 0.56	1.37 ± 0.57	1.44 ± 0.66	0.195			
Sleep quality	59.64 ± 25.11	65.13 ± 28.50	67.43 ± 27.31	73.42 ± 26.25	0.001	0.001	0.182	0.639
Anxiety	3.04 ± 2.32	3.34 ± 2.33	3.02 ± 1.93	2.91 ± 1.90	0.525			
Depression	1.97 ± 1.87	2.71 ± 1.95	2.49 ± 1.91	2.28 ± 1.93	0.018	1.000	0.721	1.000

**Table 5 ijerph-19-05889-t005:** Fourth wave. Health outcomes associated with procedural justice and occupational stress. Multivariate logistic regression model adjusted for age and gender.

Predictor	Dependent Variable aOR (CI 95%)					
	Anxious ^1^	Depressed ^2^	Burned-Out ^3^	Satisfied ^4^	Happy ^3^	Intention to Stay
Procedural justice	0.866 (0.645–1.164)	1.045 (0.824–1.325)	0.868 (0.651–1.158)	1.386 (1.032–1.863) *	1.142 (0.905–1.441)	1.030 (0.810–1.309)
Effort	1.283 (0.879–1.892)	1.268 (0.944–1.703)	1.551 (1.051–2.289) *	0.706 (0.482–1.036)	0.961 (0.733–1.259)	0.762 (0.560–1.039)
Reward	0.856 (0.718–1.020)	0.846 (0.731–0.978) *	0.880 (0.746–1.037)	1.237 (1.048–1.460) *	1.105 (0.965–1.266)	1.179 (1.023–1.358) *

Notes: ^1^ = GADS anxiety score ≥ 5; ^2^ = GADS depression score ≥ 2; ^3^ = dichotomized at the median; ^4^ = moderately, very, or extremely satisfied. * = *p* < 0.05.

## Data Availability

Data deposited on Zenodo DOI:10.5281/zenodo.6370401. The study protocol will be provided upon request.

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
