# Peer review of "Treating Anti-Vax Patients, a New Occupational Stressor—Data from the 4th Wave of the Prospective Study of Intensivists and COVID-19 (PSIC)"

_ijerph, 2022, doi:10.3390/ijerph19105889_

Round 1
Reviewer 1 Report
Search methods were well described.
First of all congratulations for the work done.
Secondly, to tell you that it is a very innovative work and therefore pertinent.
Did a library help with this?
Author Response
Search methods were well described.
First of all congratulations for the work done.
Secondly, to tell you that it is a very innovative work and therefore pertinent.
Did a library help with this?
Response: I thank the reviewer for the appreciation of our work
Reviewer 2 Report
Major revisions are required, specifically, a more detailed explanation of the methodology of this article.
Line 36 , leave the hyphen out of the word “syndrome”.
Line 84, leave the hyphen out of the word “justice”.
The paragraph “participants” should contain the number of them.
Instruments could be better described, for example by making a clearer list and also writing some examples of items, not only of standardized questionnaires but also ad of hoc questionnaires.
Line 93, leave the hyphen out of the word “description”.
The objectives of the study are not very clear, I would suggest rewriting them.
The results must be in order of appearance of the objectives.
The numbers of participants are not clear, there is confusion between Table 1 and the paragraph of the results. Describe also in the paragraph the number of participants in the 4 phases
The subjects analyzed are the same? Why the number of participants varies in the various timing? I appreciated the comparisons made by the authors but at this point could not be done a longitudinal study on the same subjects of the first peak?
Line 325, leave the hyphen out of the word “order”.
The reader could blame anti-vax people like those who stress out doctors, it’s a delicate matter. I suggest changing the title, choosing more ethical one that better described the article.
Author Response
#2
Major revisions are required, specifically, a more detailed explanation of the methodology of this article.
Line 36 , leave the hyphen out of the word “syndrome”.
Line 84, leave the hyphen out of the word “justice”.
Line 93, leave the hyphen out of the word “description”.
Line 325, leave the hyphen out of the word “order”.
Response: I thank the reviewer for pointing out some typos that were introduced in the editorial process of recomposing the manuscript by an employee of the MDPI.
The paragraph “participants” should contain the number of them.
Response. I agree. I have added the specification:
Over the course of the study, the total number of ICPs on duty in the COVID-19 hospital ranged from 155 (1st wave) to 205 (2nd wave) and 198 (3rd and 4th waves). In all the surveys, the participants constituted a significant share of those eligible.
Instruments could be better described, for example by making a clearer list and also writing some examples of items, not only of standardized questionnaires but also ad of hoc questionnaires.
R.: I gladly added a detailed description of the instrument used, which had already been done in the previous stages of the study. I also measured the value of the internal coherence (Cronbach's alpha) of the questionnaires in this present survey. I omitted these parts because they were reported in previous articles, but the reviewer is right, it is appropriate to put them to allow those who read only this article to understand. The manuscript is now as it follows:
The questionnaire included a series of ad hoc questions on occupational variations in relation to the pandemic, and a panel of standardized questionnaires measuring occupational and emotional outcomes.
The questions in the ad hoc part of the questionnaire were obtained prior to the start of the survey through structured interviews in a focus group conducted on a small group of anesthetists. Before each new data collection, the authors discussed to decide if there was the possibility of reducing the number of questions, to favor the answer and reduce the loss of observations which is a frequent problem in longitudinal studies.
The ICPs were asked to indicate the extent of their workload in the current situation compared to the past, by choosing one of 5 responses ranging from “much less than usual” to “much greater than usual”. Similarly, they were required to indicate (with the same 5-point Likert-type scale) how much time they were spending on physical activity and, respectively, meditation, prayer, or spiritual/mental activities, compared to what they did before the epidemic.
Furthermore, they were asked to indicate whether they agreed that during the epidemic their work had become more monotonous and repetitive; that the task of informing relatives of the death of a patient had been more frequent; that at work they were increasingly isolated; and that in social life they were increasingly isolated. In each of these questions, they could choose the answer from a 5-point Likert-type scale ranging from “I strongly disagree” to “I absolutely agree”.
Procedural justice perceived in safety measures was measured with the Italian version [7] of the Colquitt questionnaire [8–10] which is composed by 3 items (e.g., “Are these procedures error-free?”). Each question was answered according to a 5-point Likert scale, from 1 = “I totally disagree” to 5 = “I strongly agree”, thus producing a scale ranging from 3 to 15. In this 4th survey, the reliability of the questionnaire, measured by Cronbach’s alpha, was 0.660 (acceptable).
Work stress was assessed with the Italian version [11, 12] of the Siegrist effort/reward imbalance model [13, 14]. The questionnaire consisted of 10 items with responses ranging on a 4-point Likert scale from “1 = strongly disagree” to “4 = strongly agree”. The effort subscale contained three questions (e.g., “My job has become more and more demanding”); the total score ranged from 3 to 12. The reward sub-scale was based on seven questions (e.g., “I receive the respect I deserve from my superior or an equivalently qualified person”); consequently, this score ranged from 7 to 28. Internal consistency reliability in this survey was 0.798 for effort (very good) and 0.823 for reward (very good).
Sleep quality was determined using the 2-item version of the “Sleep Condition Indicator” (SCI-02) [15, 16], a brief scale that evaluates insomnia disorder in everyday clinical practice, according to the Diagnostic Statistic Manual 5 (DSM5). Each question (e.g., “How many nights a week have you had a problem with your sleep during the past month?”) was graded on a 5-point Likert scale, ranging from 4 to 0. The final score ranged between 0 and 8, with higher values indicating better sleep quality. Cronbach’s alpha was 0.759 (good). A score of ≤4 revealed possible insomnia disorder.
Mental health was measured on Goldberg's anxiety and depression scales (GADS) [17, 18], composed of 18 binary items on anxiety (9 items) and depression (9 items). Typical questions were: “Have you had difficulty relaxing?” for anxiety, and “Have you lost confidence in yourself?” for depression.
Other questionnaires were: job satisfaction, expressed according to Warr et al. [19, 20] by a single question on a 7-point Likert scale ranging from extremely dissatisfied to extremely satisfied; happiness, measured by the 10-point Ab-del-Khalek’s single item scale [21]; burnout feelings, evaluated according to West et al. [22] on a 6-point scale, and the intention to quit the hospital, measured with a single item (yes/no). A detailed description of the questionnaires and their psychometric properties has been published in previous reports [6, 23, 24]. In this fourth survey, the relationship with patients belonging to the anti-vaccination movement was investigated using the Italian version [25] of the "contact with patients and their family" sub-scale of the Nurses Work Functioning Questionnaire NWFQ [26]. This scale is made up of 8 questions, with answers graded from 0 to 6; the score therefore varies between 0 and 48. An example of a question is: “In the last 4 weeks, how much difficulty have you had in interacting either with patients who re-fuse the anti-COVID19 vaccination or with their relatives/careers?”. The reliability of the questionnaire (Cronbach’s alpha) in this study was 0.875.
The objectives of the study are not very clear, I would suggest rewriting them.
R.: We welcomed this suggestion and rewrote the aims section, first indicating the general purpose of the prospective study (analyzing changes in mental health of anesthesiologists following the pandemic) and then the theme that was added in the fourth wave (the reports of doctors with patients of the anti-vaccination movement). The text now is as it follows:
In this fourth survey of the PSIC we continued the analysis of the changes in the mental health of doctors who had been treating COVID-19 patients since the beginning of the pandemic, in relation to changing environmental and working conditions. Moreover, we wanted to evaluate the extent to which the relationship with anti-vax patients generated occupational stress in intensivists.
Th e results must be in order of appearance of the objectives. The numbers of participants are not clear, there is confusion between Table 1 and the paragraph of the results. Describe also in the paragraph the number of participants in the 4 phases
R.: I am sorry to note that some typos have appeared in the editorial review of the manuscript. In addition to the words with a hyphen that were not in the version submitted by us, a line was skipped in the results and a paragraph was composed as a caption for a table. The text was incomprehensible. We promptly re-established the correct graphics and are confident that the manuscript is now legible.
The number of participants in the fourth survey is reported in the first line of the results. The number of participants in the various surveys was reported in the first line of Table 1; now we added in the Table a line with the number of eligible persons.
The subjects analyzed are the same? Why the number of participants varies in the various timing? I appreciated the comparisons made by the authors but at this point could not be done a longitudinal study on the same subjects of the first peak?
R.: The longitudinal study involved a mobile cohort. Doctors could join the cohort for hiring or transfer to the COVID-19 hospital from other parts of the university or they could leave the cohort for dismissal, retirement, transfer to another hospital. The participation was anonymous. Consequently, from an epidemiological point of view, the study was a repeated cross-sectional, in which it was possible to measure and compare the punctual prevalence during the four pandemic phases, but not to calculate the incidence.
The reader could blame anti-vax people like those who stress out doctors, it’s a delicate matter. I suggest changing the title, choosing more ethical one that better described the article.
R.: We agree with the reviewer that the doctor-patient relationship is a sensitive issue and, in this case, the relationship between the doctor treating a disease and those who oppose the officially recognized treatment of the disease is particularly sensitive. The reviewer is right. The title was not informative enough. We have modified in: Treating Anti-Vax Patients, a New Occupational Stressor. Data From the 4th wave of the Prospective Study of Intensivists and COVID-19 (PSIC).

Reviewer 3 Report
This paper describes a topic that would be interesting to many readers. However, I have to say the construct of your manuscript is not well designed, and the writing is not so clear. Below are some of the major points for your reference.
- Your writing seems emphasizing your findings from the 4th wave of data collection. However, you also have an objective to assess the changes from wave 1 to wave 4. If that is the case, which is fine, you could organize your writing accordingly to avoid confusion.
- In your methodology section, under participants, only the 4th wave participants were described.
- Under the questionnaire, it seems all questionnaires used in wave 1 to wave 4 were described
- Under the statistics, it would be better to organize your writing to reflect your analysis strategies to the study for the 4th wave data, and the study for the change across the four waves. You mentioned using multiple logistic regression for the changes. I figure that is how you analyze the four-wave data
- Your result was not well presented. The first paragraph seems to describe your samples in the 4th wave. Moving down to the second paragraph I start to get lost.
- Line 131, you started with “Table 19. patients (44.2%) was significantly higher than at baseline (24.7%) but was”. I don’t know what you are talking about.
- Table 1 again is the characteristic of study population but from all four waves
- Immediate after Table 1 is the impact of changes along with table 2. What does the change mean? From wave 1 to wave 4?
- For your next Table 3, you did present statistics for all four waves. But I cannot figure what the percentages mean. I cannot find answers from your narratives. For example, you stated, “The percentage of workers 163 suffering from excessive stress did not change significantly”. I cannot find an item in your Table 3 that is about excessive stress.
Similar writing issues can be seen in the remain text. I feel you really want to re-organize your design as what a story you want to tell, and write out in a clearer way.
Author Response
#3
This paper describes a topic that would be interesting to many readers. However, I have to say the construct of your manuscript is not well designed, and the writing is not so clear. Below are some of the major points for your reference.
Your writing seems emphasizing your findings from the 4th wave of data collection. However, you also have an objective to assess the changes from wave 1 to wave 4. If that is the case, which is fine, you could organize your writing accordingly to avoid confusion.
Response: We are indebted to the reviewer for the fact that he / she has grasped the interesting aspects of our work and took his time to allow us to improve it. According to this suggestion, we have rewritten the aims section, first indicating the general purpose of the prospective study (analyzing changes in mental health of anesthesiologists following the pandemic) and then the theme that was added in the fourth wave, the relationship with anti-vaccination patients.
In your methodology section, under participants, only the 4th wave participants were described.
Response: Accepting the reviewer's suggestion, we indicated in the participants section the size of the cohort during the different phases of the pandemic. The number of participants in each survey was formerly reported in the first line of Table 1; we have now added another line indicating the total number of ICPs working in the hospital during each survey (eligible persons).
Under the questionnaire, it seems all questionnaires used in wave 1 to wave 4 were described.
Response: Accepting the reviewer's suggestion, we provided more information on all the questionnaires that were used in the different phases of the study. In the previous version of the manuscript, I had omitted these parts because they were reported in previous articles, but the reviewer is right, it is appropriate to put them to allow those who read only this article to understand. The manuscript now is as it follows:
The questionnaire included a series of ad hoc questions on occupational variations in relation to the pandemic, and a panel of standardized questionnaires measuring occupational and emotional outcomes.
The questions in the ad hoc part of the questionnaire were obtained prior to the start of the survey through structured interviews in a focus group conducted on a small group of anesthetists. Before each new data collection, the authors discussed to decide if there was the possibility of reducing the number of questions, to favor the answer and reduce the loss of observations which is a frequent problem in longitudinal studies.
The ICPs were asked to indicate the extent of their workload in the current situation compared to the past, by choosing one of 5 responses ranging from “much less than usual” to “much greater than usual”. Similarly, they were required to indicate (with the same 5-point Likert-type scale) how much time they were spending on physical activity and, respectively, meditation, prayer, or spiritual/mental activities, compared to what they did before the epidemic.
Furthermore, they were asked to indicate whether they agreed that during the epidemic their work had become more monotonous and repetitive; that the task of informing relatives of the death of a patient had been more frequent; that at work they were increasingly isolated; and that in social life they were increasingly isolated. In each of these questions, they could choose the answer from a 5-point Likert-type scale ranging from “I strongly disagree” to “I absolutely agree”.
Procedural justice perceived in safety measures was measured with the Italian version [7] of the Colquitt questionnaire [8–10] which is composed by 3 items (e.g., “Are these procedures error-free?”). Each question was answered according to a 5-point Likert scale, from 1 = “I totally disagree” to 5 = “I strongly agree”, thus producing a scale ranging from 3 to 15. In this 4th survey, the reliability of the questionnaire, measured by Cronbach’s alpha, was 0.660 (acceptable).
Work stress was assessed with the Italian version [11, 12] of the Siegrist effort/reward imbalance model [13, 14]. The questionnaire consisted of 10 items with responses ranging on a 4-point Likert scale from “1 = strongly disagree” to “4 = strongly agree”. The effort subscale contained three questions (e.g., “My job has become more and more demanding”); the total score ranged from 3 to 12. The reward sub-scale was based on seven questions (e.g., “I receive the respect I deserve from my superior or an equivalently qualified person”); consequently, this score ranged from 7 to 28. Internal consistency reliability in this survey was 0.798 for effort (very good) and 0.823 for reward (very good).
Sleep quality was determined using the 2-item version of the “Sleep Condition Indicator” (SCI-02) [15, 16], a brief scale that evaluates insomnia disorder in everyday clinical practice, according to the Diagnostic Statistic Manual 5 (DSM5). Each question (e.g., “How many nights a week have you had a problem with your sleep during the past month?”) was graded on a 5-point Likert scale, ranging from 4 to 0. The final score ranged between 0 and 8, with higher values indicating better sleep quality. Cronbach’s alpha was 0.759 (good). A score of ≤4 revealed possible insomnia disorder.
Mental health was measured on Goldberg's anxiety and depression scales (GADS) [17, 18], composed of 18 binary items on anxiety (9 items) and depression (9 items). Typical questions were: “Have you had difficulty relaxing?” for anxiety, and “Have you lost confidence in yourself?” for depression.
Other questionnaires were: job satisfaction, expressed according to Warr et al. [19, 20] by a single question on a 7-point Likert scale ranging from extremely dissatisfied to extremely satisfied; happiness, measured by the 10-point Ab-del-Khalek’s single item scale [21]; burnout feelings, evaluated according to West et al. [22] on a 6-point scale, and the intention to quit the hospital, measured with a single item (yes/no). A detailed description of the questionnaires and their psychometric properties has been published in previous reports [6, 23, 24]. In this fourth survey, the relationship with patients belonging to the anti-vaccination movement was investigated using the Italian version [25] of the "contact with patients and their family" sub-scale of the Nurses Work Functioning Questionnaire NWFQ [26]. This scale is made up of 8 questions, with answers graded from 0 to 6; the score therefore varies between 0 and 48. An example of a question is: “In the last 4 weeks, how much difficulty have you had in interacting either with patients who re-fuse the anti-COVID19 vaccination or with their relatives/careers?”. The reliability of the questionnaire (Cronbach’s alpha) in this study was 0.875.
Under the statistics, it would be better to organize your writing to reflect your analysis strategies to the study for the 4th wave data, and the study for the change across the four waves. You mentioned using multiple logistic regression for the changes. I figure that is how you analyze the four-wave data
Response: The reviewer got the point of the work exactly and helped us improve readers’ understanding. According to the repeated cross-sectional design, we calculated the punctual frequency of the variables of interest and compared this prevalence of the 4th survey with those collected in the previous pandemic phases. This comparison was made with Student’s t or chi square statistics and is reported in Table 1. Then we tried to understand which were the most important stressors in the 4th phase. This was done by logistic regression. We added the specification that “We therefore wanted to verify whether, even in this fourth phase, environmental and occupational factors had the same effect on stress and mental health as had been observed on previous occasions.” The text now is as it follows:
The distribution of the variables was analyzed by measuring central tendency (mean, median, mode) and dispersion (standard deviation). According to the repeated cross-sectional design, we calculated the punctual frequency of the variables of interest and compared this prevalence of the 4th survey with those collected in the previous pandemic phases by the chi-square test for categorical data or by ANOVA and post-hoc comparison according to Bonferroni for continuous variables.
We therefore wanted to understand which were the most important occupational stressors in the 4th phase. The effect of changes in work patterns on occupational stress was studied using multiple logistic regression, in which age, gender and all changes were included as predictors, and ERI, dichotomized using 1 as the cut-off, was entered as the dependent variable. In this way it was possible to calculate the adjusted odds ratio and the 95% confidence intervals for each of the pandemic changes.
Similarly, the effect on health outcomes due to perception of occupational justice, effort, and reward was studied using multiple logistic regression in which age and gender were postulated as confounders and the three work-related variables as predictors.
Analyses were performed using IBM/SPSS 26.0 (IBM Corporation, Armonk, NY, USA).
Your result was not well presented. The first paragraph seems to describe your samples in the 4th wave. Moving down to the second paragraph I start to get lost.
Response: The reviewer is right, going from the first to the second paragraph of the manuscript there was a jump. The difficulty of comprehension was increased by a compositional error introduced by the MDPI employees. We have clarified in the methods section that, since the cohort is mobile and the survey is anonymous, the study has a repeated cross-sectional character, which allows a comparison of the point prevalence measured in the different surveys, but we cannot calculate the incidence. For this reason, we compared the prevalence of the different responses found in the 4th survey with those collected previously.
Line 131, you started with “Table 19. patients (44.2%) was significantly higher than at baseline (24.7%) but was”. I don’t know what you are talking about.
Table 1 again is the characteristic of study population but from all four waves
R.: I am very sorry to have noticed that some typos have appeared in the editorial review of the manuscript. In addition to the words with a hyphen that were not in the version submitted by us, a line was skipped in the results and a paragraph was composed as a caption for a table. I have been collaborating for many years with MDPI journals where I have published 23 articles, and this has never happened before. I promptly re-established the correct graphics and I’m confident that the manuscript is now legible.
Immediate after Table 1 is the impact of changes along with table 2. What does the change mean? From wave 1 to wave 4?
R.: I thank the reviewer who pointed out an aspect that I missed. The sentence was not clear enough. We have now edited the text making it clear that we are talking about “The impact of changes in work patterns induced by the pandemic on occupational stress perceived during the 4th survey”.
For your next Table 3, you did present statistics for all four waves. But I cannot figure what the percentages mean. I cannot find answers from your narratives. For example, you stated, “The percentage of workers 163 suffering from excessive stress did not change significantly”. I cannot find an item in your Table 3 that is about excessive stress.
R.: The percentage of distressed workers is shown in Table 3. I corrected the text by replacing "excessive stress" with "distressed".
Similar writing issues can be seen in the remain text. I feel you really want to re-organize your design as what a story you want to tell, and write out in a clearer way.
R.: I thank the reviewer for the careful review. We carefully checked the manuscript, looking for other points where the speech was not too clear and correcting them.

Reviewer 4 Report
Thank you very much for the article, which shows important insights into the encounter of medical staff with anti-vaccination activists in the field of work.
However, from my point of view, the article has weaknesses in terms of the stringency of the content and the methodology reported, so the article should be heavily revised again before publication.
In general, please consider focusing the content of the article and concentrate either on the report of all four waves or on the report of the results from wave 4. Jumping back and forward between the results from four waves and the results only from wave 4 unfortunately confuses the reader. Please use the STROBE statement as a guide when reviewing and writing the text.
https://www.strobe-statement.org/download/italian-translation-of-strobe-statement
Please also check the text for existing hyphenations, which are probably still present in some words due to automatic hyphenation in the text.
The Conclusion and Discussion are well written, although more explicit reference could be made to the results presented in the tables. No reference is made to, for example, the model quality and thus to what extent the impact of changes in work patterns on occupational stress is explained by the model. My following comments concentrate on the first part of the article.
Headline: It is not clear from the headline that explicit results from wave 4 are also reported here. In addition, "doctors" are referred to here. Furthermore, it is not always clear from the text that this is the case, as except for the introduction, it is referred to as health care workers or workers.
Line 62: It is not clear from the previous paragraphs that these are reported results from the first three surveys of the PSIC study. This should be made clear here and the introduction of the study should be placed at the beginning. Especially since reference is made to all 4 waves in the following. The term "doctors" is also used here for the last time up to the discussion. In my opinion, health care workers and doctors or physicians are not congruent.
Line 70: "workers" ? Even if it becomes more concrete under results and health care workers are certainly meant, it should be described here in more detail who was contacted. Or whether individual occupational groups were deliberately left out. So only "Nurses" or also "Physicians" etc.? Or "all HCW" or only "doctors"?
Line 75: Was recruitment only at baseline, or was there also follow-up recruitment? How many participants took part across all 4 waves?
Line 81: The methodology is missing, how was the survey implemented? Paper/pencil questionnaire, online survey?
Line 123: Only inclusion/exclusion from wave 4 mentioned here due to incomplete responses. What about other drop-outs? For the sake of completeness, this should also be reported for waves 1-3. Information should also be provided on the drop-outs. A flowchart would be appropriate here.
In addition, when comparing the four cross-sectional results, distortions can be assumed due to the varying number of participants. Changes across the four waves are thus not clearly comprehensible. Why a longitudinal data set wasn’t created that only included participants in all four waves? If only anonymous respondents were interviewed, from my point of view statements about changes over the course of the four waves can only be made to a limited extent, since it is not clear whether the same respondents always answered. Or were pseudonyms used?
Line 128: Qualified specialists are now mentioned here. A clear subdivision of the occupational groups is appropriate here.
Line 131: Table 19? The table assignment is wrong here, as there is the explanatory text for table 1.
Line 145: It must be made clear here that these are four reported cross-sectional results.
Table 1: Especially with the "reporting long-COVID" are these different participants in wave 4 to wave 3, or is this a repeat statement by the same person?
This should be made clear. In addition, it should be considered whether it makes sense to divide the participant data into participants who participated across all four waves and those who did not participate consistently. The mixing of percentages in each wave from all participants as well as from subpopulations is also not well resolved. Please differentiate better. Example: Wave 3: 68/120 participants (resident) = 56.7% but 10/120 participants (reporting long-COVID) = 38.5%, which is not easy to understand from the frequencies given (100% would be 26 people). What is the denominator there? Is this a transmission error in the N=23 "Reporting COVID-19 disease"?
Table 2: Only the results of wave 4 are shown in Table 2. This must be stated here. In addition, the sample size and the frequency per variable should also be given here.
Table 3: Here, too, the total population of the wave must be given for the sake of completeness. Changes are only of limited significance due to varying numbers of participants and the associated characteristics.
Author Response
#4
Thank you very much for the article, which shows important insights into the encounter of medical staff with anti-vaccination activists in the field of work.
However, from my point of view, the article has weaknesses in terms of the stringency of the content and the methodology reported, so the article should be heavily revised again before publication.
In general, please consider focusing the content of the article and concentrate either on the report of all four waves or on the report of the results from wave 4. Jumping back and forward between the results from four waves and the results only from wave 4 unfortunately confuses the reader. Please use the STROBE statement as a guide when reviewing and writing the text.
https://www.strobe-statement.org/download/italian-translation-of-strobe-statement
Please also check the text for existing hyphenations, which are probably still present in some words due to automatic hyphenation in the text.
Response: We appreciated that the reviewer spent so much time reviewing the manuscript and made suggestions that were helpful to us. This observation is very stimulating and led us to reorganize the text. The problem reported by the reviewer was real and depended on two factors: one, that trying to summarize the work done, we were sometimes not very understandable. The second, that the reading of the article was unfortunately made difficult by a series of typos and omissions that occurred during the editorial process and were present in the manuscript sent to the reviewers, but not in the original submitted by us. I am very sorry to have noticed that many words have appeared with a hyphen, a line was skipped in the results and a paragraph was composed as a caption for a table. I have been collaborating for many years with MDPI journals where I have published over 20 articles, and this has never happened before. I promptly re-established the correct graphics and I’m confident that the manuscript is now legible.
Regarding the methodology, we have included some data that we had previously omitted because they have already been reported in previous articles. In particular, we have indicated the number of eligible people. We have detailed the characteristics of the tool used for data collection. We reorganized the results by indicating the titles of the different topics covered in the sub-sections. In carrying out the research, in addition to the STROBE statement that the reviewer kindly pointed out to us, we adhered to the ASPIRE guidelines for repeated cross-sectional studies [Gryaznov D, Odutayo A, von Niederhäusern B, et al. Rationale and design of repeated cross-sectional studies to evaluate the reporting quality of trial protocols: the Adherence to SPIrit REcommendations (ASPIRE) study and associated projects. Trials. 2020 Oct 28;21(1):896. doi: 10.1186/s13063-020-04808-y.].
The Conclusion and Discussion are well written, although more explicit reference could be made to the results presented in the tables. No reference is made to, for example, the model quality and thus to what extent the impact of changes in work patterns on occupational stress is explained by the model. My following comments concentrate on the first part of the article.
Response .: I am grateful for this observation which allowed us to deal with the presentation of the results in greater detail. We have added some paragraphs that explain the results in more detail and link them with those obtained in previous surveys. We added the following:
The stress model we investigated referred exclusively to work factors that have changed during the pandemic. We have not considered the intrinsic stressors in the work of anesthesiologist and more generally those of which no variation has been reported to us. Taking this into consideration, the model shown in Table 2 appears to effectively explain the observed increase in distress cases.
In correspondence with the fourth pandemic wave, the increase in cases of distress was mainly due to monotony, isolation, difficulty in relating with anti-vax patients and lack of time for meditation. In previous investigations, others were the dominant stressors. At baseline, at the first stage, stress was mainly driven by unprotected exposures to COVID-19 positive patients and lack of confidence in safety measures; compassion fatigue was nearly significant [23]. These findings perfectly expressed the situation of the first pandemic phase, in which the ICPs had to deal with an unknown disease, against which there was a lack of safety measures and therapies. The situation has gradually changed. After the second wave, the most relevant stressors, besides the distrust in security measures and excessive workload, were the lack of time for meditation and isolation [6]. Working alone and with little confidence in safety measures were the most significant predictors of distress in the third wave [24]. Confidence in safety measures is not a major stressor today, probably because ICPs have assimilated the safety procedures imposed by the pandemic and observed that COVID-19 infection often originates from non-occupational exposure.
In this fourth wave occupational stress has a modest association with depression, burnout, job satisfaction and intention to stay, and no relation with anxiety and happiness. A few months ago, during the third phase, occupational effort and lack of rewards were very significantly associated with anxiety, depression, burnout, job dissatisfaction, and intention to quit [24]. This attenuation of the link between occupational stress caused by the pandemic and negative emotional outcomes could indicate that ICPs are moving from emergency to the routine management of an endemic clinical problem. Acute distress is turning into chronic distress.
Headline: It is not clear from the headline that explicit results from wave 4 are also reported here. In addition, "doctors" are referred to here. Furthermore, it is not always clear from the text that this is the case, as except for the introduction, it is referred to as health care workers or workers.
Response: The reviewer is right. The title was not informative enough. We have modified in: “Treating Anti-Vax Patients, a New Occupational Stressor. Data From the 4th wave of the Prospective Study of Intensivists and COVID-19 (PSIC).”
We tried to make it clear in the manuscript that we studied anesthetists-intensivists. Consequently, we have replaced the specific term in cases where it generically referred to health workers.
Line 62: It is not clear from the previous paragraphs that these are reported results from the first three surveys of the PSIC study. This should be made clear here and the introduction of the study should be placed at the beginning. Especially since reference is made to all 4 waves in the following. The term "doctors" is also used here for the last time up to the discussion. In my opinion, health care workers and doctors or physicians are not congruent.
R.: Welcoming the suggestion, we explained in the first paragraph of the introduction that this is a prospective study on intensivists. To avoid difficulties, we have divided the Results into sections.
Line 70: "workers" ? Even if it becomes more concrete under results and health care workers are certainly meant, it should be described here in more detail who was contacted. Or whether individual occupational groups were deliberately left out. So only "Nurses" or also "Physicians" etc.? Or "all HCW" or only "doctors"?
R.: This observation is correct. We have corrected these references throughout the text
Line 75: Was recruitment only at baseline, or was there also follow-up recruitment? How many participants took part across all 4 waves?
Line 81: The methodology is missing, how was the survey implemented? Paper/pencil questionnaire, online survey?
R.: The reviewer correctly reported that the "participants" section was missing some important information. We have added two paragraphs to say that:
Repeated baseline surveys were carried out to correspond with the first, second (December 2020), third (April 2021) and recent fourth wave of the pandemic (December 2021). On this occasion, as in the previous ones, the ICPs were confidentially contacted by email and asked to participate through the SurveyMonkey© online platform. The answers were collected anonymously on a specific file without any individual reference. Participation was completely voluntary, and no economic incentive was provided for response. Two weeks after commencing the investigation, a reminder mail was sent with the preliminary results of the study. The ICPs were informed both of the results collected in the various surveys and the findings published in scientific journals.
Since the cohort was mobile and the survey was anonymous, the prospective study had a repeated cross-sectional character, which allows a comparison of the point prevalence measured in the various surveys but does not allow for the calculation of the incidence.
Over the course of the study, the total number of ICPs on duty in the COVID-19 hospital ranged from 155 (1st wave) to 205 (2nd wave) and 198 (3rd and 4th waves). In all the surveys, the participants constituted a significant share of those eligible.
Line 123: Only inclusion/exclusion from wave 4 mentioned here due to incomplete responses. What about other drop-outs? For the sake of completeness, this should also be reported for waves 1-3. Information should also be provided on the drop-outs. A flowchart would be appropriate here.
R.: There were no other reasons for exclusion, in this one as in the previous surveys, other than the incompleteness of the answers. The system used (SurveyMonkey) does not allow resuming the questionnaire when it is interrupted. In the first waves, incomplete responses were sporadic, while in this latest survey they were frequent. In the manuscript we explained that:
Fifteen gave incomplete answers and were therefore excluded from subsequent analyses. This high number of interruptions before the conclusion of the questionnaire was a new phenomenon; in previous surveys there had been only a few sporadic cases. In intensivists, the interruption of a questionnaire may depend on the need to respond to a medical emergency. In this case, we fear that one reason might be the boredom of answering the same questions for the fourth time.
In addition, when comparing the four cross-sectional results, distortions can be assumed due to the varying number of participants. Changes across the four waves are thus not clearly comprehensible. Why a longitudinal data set wasn’t created that only included participants in all four waves? If only anonymous respondents were interviewed, from my point of view statements about changes over the course of the four waves can only be made to a limited extent, since it is not clear whether the same respondents always answered. Or were pseudonyms used?
Response: The limited size of the population advised to guarantee anonymity by avoiding codes and pseudonyms that do not give sufficient guarantees in a small group. The cohort was also mobile, that is, although the number of doctors remained stationary, there was a turnover. For these reasons, we have decided to follow a repeated cross-sectional and not longitudinal design. In all surveys, the participants were a significant share of the population.
Line 128: Qualified specialists are now mentioned here. A clear subdivision of the occupational groups is appropriate here.
R.: There were only two groups of doctors in the population: specialists, who had permanent employment, and last-year residents who were hired on a fixed-term contract for the pandemic. The dimensions of the two categories in the different waves are shown in table 1.
Line 131: Table 19? The table assignment is wrong here, as there is the explanatory text for table 1.
Response: Sorry, this is a typo introduced by MDPI.
Line 145: It must be made clear here that these are four reported cross-sectional results.
R.: I have completed the table legend “Characteristics of the population in the different phases of the prospective study.”
Table 1: Especially with the "reporting long-COVID" are these different participants in wave 4 to wave 3, or is this a repeat statement by the same person?
R.: Sorry, we can't know. It is not possible to know who has answered only one, none or all of the surveys.
This should be made clear. In addition, it should be considered whether it makes sense to divide the participant data into participants who participated across all four waves and those who did not participate consistently. The mixing of percentages in each wave from all participants as well as from subpopulations is also not well resolved. Please differentiate better. Example: Wave 3: 68/120 participants (resident) = 56.7% but 10/120 participants (reporting long-COVID) = 38.5%, which is not easy to understand from the frequencies given (100% would be 26 people). What is the denominator there? Is this a transmission error in the N=23 "Reporting COVID-19 disease"?
R.: I am grateful to the reviewer for careful checking that allowed me to find an error in the table. In the third wave, in fact, those who reported having had COVID-19 were 10 + 14 + 2 = 26, not 23. To avoid confusion about the percentages we have inserted a horizontal line in the table and have indicated next to each variable to whom the percentage refers. The percentages at the top of the table refer to the participants, at the bottom the percentages of severity of the disease refer to the subjects with the disease.
Table 2: Only the results of wave 4 are shown in Table 2. This must be stated here. In addition, the sample size and the frequency per variable should also be given here.
Response: The table legend has been changed, as suggested, to: “Impact of changes in work patterns on occupational stress (ERI> 1) during the 4th pandemic phase. Odds ratios (aOR) and Confidence Intervals at 95% (CI95%) adjusted for age and sex. 95 cases.” Mean and standard deviation have been added next to each variable.
Table 3: Here, too, the total population of the wave must be given for the sake of completeness. Changes are only of limited significance due to varying numbers of participants and the associated characteristics.
Response: We agree with the reviewer. The population size is small, and this is a limitation of the study we discussed. The population from which the samples of the four waves are taken is shown in table 1; it remained numerically stable. We agree that changes are only of limited significance due to the characteristic of the study.

Round 2
Reviewer 2 Report
Thank Author for your edit to article. Now is more readable and clear.
Author Response
We all sincerely thank the reviewer for the attention with which he / she has considered our work. His / her suggestions were very helpful in improving the manuscript
Reviewer 3 Report
I think the authors made great progress in revising or modifying the manuscript. It is much clear as what this paper is to present and focus on. The changes made are substantial.
Author Response

(The authors gave the same response as above.)

Reviewer 4 Report
Thank you for the extensive revision of the manuscript and the detailed response to my comments.
The text reads much better now and is more comprehensible.
I don't know if it's due to the change mode of the revised version, but in the new parts of the text, words are often displayed twice or spaces are missing. For example, in line 50: ICPsICP. Please proofread the text again.
Otherwise, just a small comment on the content of Table 4: In lines 253-254 you wrote "...Bonferroni test which was used to compare the fourth and first waves." The p-values 4 vs. 2 are missing from the table.
Author Response
The reviewer is right. In the editorial version, which is different from the one submitted by us, there are many words repeated. We will be particularly careful with the final proofs to avoid these typos.
I added in Table 4 the comparisons (4 vs.2) made with the Bonferroni test, which were all insignificant.
We thank the reviewer for the attention he / she has contributed to improving our manuscript